# The Role of Psychological Wellbeing in a Cross-Provider Worksite Healthcare Management Program for Employees with Musculoskeletal Disorders

**DOI:** 10.3390/ijerph19095452

**Published:** 2022-04-29

**Authors:** Lara Lindert, Lara Schlomann, Holger Pfaff, Kyung-Eun (Anna) Choi

**Affiliations:** 1Institute of Medical Sociology, Health Services Research and Rehabilitation Science, Faculty of Medicine, University Hospital Cologne, Faculty of Human Sciences, University of Cologne, Eupener Str. 129, 50933 Cologne, Germany; lara.schlomann@uk-koeln.de (L.S.); holger.pfaff@uk-koeln.de (H.P.); anna.choi@mhb-fontane.de (K.-E.C.); 2Center for Health Services Research, Brandenburg Medical School Theodor Fontane, Fehrbelliner Str. 38, 16816 Neuruppin, Germany

**Keywords:** disability score, pain intensity, prevention, occupational health, mental wellbeing

## Abstract

Background: Musculoskeletal and mental disorders are often comorbid, with complex correlations of pain, impairment due to pain, disability, and psychological wellbeing. This study investigates the role of psychological wellbeing in a worksite healthcare program for employees within a German randomized controlled trial. Methods: For our analyses we used data of the module for minor musculoskeletal complaints (N = 180). The intervention included a workplace-related training and case manager support. Results: Changes over time were significant in the disability score (*t*(179) = 9.04, *p* < 0.001), pain intensity (*t*(179) = 9.92, *p* < 0.001), and psychological wellbeing (*t*(179) = −4.65, *p* < 0.001). Individuals with low vs. high psychological wellbeing showed significant differences in their disability score_t0,t1_ (*t*_t0_(178) = −4.230, *p*_t0_ < 0.001, *t*_t1_(178) = −2.733, *p*_t1_ < 0.001), pain intensity_t0,t1_ (*t*_t0_(178) = −3.127, *p*_t0_ < 0.01, *t*_t1_(178) = −3.345, *p*_t1_ < 0.01, and motivation_t0_ (*t*_t0_(178) = 4.223, *p*_t0_ < 0.001). The disability score_∆_ mediates the impact of pain intensity_∆_ on psychological wellbeing_t1_ (beta = 0.155, *p* < 0.05). Psychological wellbeing_∆_ had an impact on the disability score_t1_ (beta = −0.161, *p* < 0.01). Conclusions: The lower the psychological wellbeing is at an intervention’s beginning, the higher the potential is for its improvement, which might affect individuals’ experienced impairment due to pain. In order to achieve the best outcomes, interventions should include both pain-related and psychological aspects. Future research needs to explore the causality of the found interrelationships further.

## 1. Introduction

In Germany, musculoskeletal disorders (MSDs), followed by mental disorders (MDs), are the two leading causes of the absence of workdays [1] and cause high economic costs. Regarding mental and physical health, one relevant setting is the workplace: The relationship between psychological wellbeing and psychosocial work factors (e.g., social support, decision latitude, and workload) is already well-studied, and the research allows us to state that psychosocial work factors have an impact on the psychological wellbeing of employees [2,3,4]. Also, psychosocial work factors are associated with the MSD of employees [5,6,7,8,9]. 

There are several study results focusing on the relationship between psychological wellbeing and MSDs. MSDs are associated with poorer (subjective) wellbeing, and the overall psychological wellbeing is lower in groups with MSDs than in groups without them. Also, MSDs are more prevalent in workers with lower psychological wellbeing [10,11,12]. In a two-year follow-up study by Larsson et al. [13], positive psychological wellbeing predicted lower severe pain in the general Swedish population, regardless of whether persons were affected by chronic pain at the baseline or not. Regarding the role of psychological wellbeing in the context of healthcare utilization among patients with MSDs, a recent study revealed that hospital contacts are predicted by higher levels of pain and low psychological wellbeing [14]. Furthermore, Müller-Schwefe and Überall [15], e.g., state that impairment due to pain is of more importance for the quality of life than the absolute pain intensity of individuals with chronic pain. Abas et al. [16] revealed that, in older people, the relationship between impairment due to pain (the pain was interfering with individuals’ function) and psychological wellbeing was explained by disability (being not able/limited in everyday life). When it comes to interventions, a systematic review and meta-analysis found that psychological treatments have a small positive effect on the return to work for patients with MSDs and/or common MDs [17]. Furthermore, physiotherapy in patients with neck, shoulder, or low back pain leads not only to improvements in their pain but also in their psychological wellbeing [14]. 

This study focuses on data of a worksite healthcare (WHC) program for employees with MSDs in Germany and examines the role of psychological wellbeing. Even though the relationship between MSDs and mental health is already examined in different settings, there are (to our knowledge) no studies yet on the role of psychological wellbeing in a preventive WHC program for employees with first minor complaints concerning their musculoskeletal system. Most of the research, so far, focusses on individuals with chronic pain. Early detection and treatment of affected persons are particularly important, as incapacity to work and early retirement caused by MSDs are extremely frequent and cost-intensive. High physical workloads are still relevant in today’s working world. Alternatively, the increase in sedentary work leads to under-stressing of the musculoskeletal system, which may cause injuries and overstrain already due to everyday stresses. Due to the high socio-political importance of MSDs, occupational prevention is urgently required [18].

The aim of this study is to investigate the role of psychological wellbeing in a WHC program that is aimed at employees with MSDs in Germany in different company settings. For our analyses, we focused on pain intensity, employees’ experienced impairment due to pain (the disability score), and psychological wellbeing.

As physiotherapy in patients with neck, shoulder, or low back pain leads not only to improvements in pain but also in psychological wellbeing [14], and since the intervention for our sample provides regular and work-related training, we hypothesized that:

**Hypothesis** **1** **(H1).**
*The psychological wellbeing of employees increases after the intervention.*


As MSDs are associated with MDs and poorer (subjective) wellbeing, and since the overall psychological wellbeing is lower in groups with MSDs than in groups without them, and based on the fact that MSDs are more prevalent in workers with lower psychological wellbeing [10,11,12], we hypothesized that:

**Hypothesis** **2** **(H2).**
*Pain intensity is higher in individuals with lower psychological wellbeing than in individuals with higher psychological wellbeing.*


As the study results have shown that impairment due to pain is of more importance for the quality of life than absolute pain intensity [15] and that the relationship between impairment due to pain and psychological wellbeing was explained by disability [16], we further hypothesized that:

**Hypothesis** **3** **(H3).**
*The difference in disability scores (disability score*
_∆_
*) mediates the impact of the difference in pain intensity (pain intensity*
_∆_
*) on psychological wellbeing after the intervention (psychological wellbeing_t_*
_1_
*).*


As the study results have found that psychological wellbeing is of predictive value for pain intensity [13], we further examined the impact of psychological wellbeing on the disability score and hypothesized that:

**Hypothesis** **4** **(H4).**
*The difference in psychological wellbeing (psychological wellbeing*
_∆_
*) has an impact on the disability score after the intervention (disability score_t_*
_1_
*).*


## 2. Materials and Methods

This study focusses on employees with MSDs who were assigned to a workplace-related and cross-provider healthcare program in Germany. The program was part of a broader randomized controlled trial (RCT) that was conducted from April 2017 to March 2021. Twenty-two German companies (mainly steel and metal manufacturing, automotive industry, and trade and service), 12 pension funds, and 15 company health insurance funds were part of the study network [19]. The RCT included two study arms—the case management group (intervention group) and self-management group (control group)—and three modules in each study arm—module A, module B, and module C (see Figure 1). Module A was aimed at employees who had first minor complaints regarding their musculoskeletal system. Module B included employees with medium complaints who were suitable for rehabilitation. Module C was for employees with severe complaints (job is in danger and reintegration necessary). Interventions in the case management group depended on the module and included a work-related training program, rehabilitation, or psychological assessment for further action (e.g., gradual reintegration). Participants in the case management group received thorough work-related diagnostics and support from the case managers of the company’s health insurance funds. Treatment in the self-management group was oriented towards standard care, as participants received tailored information on their possibilities in regular health care [19].

Participants were recruited from August 2017 to February 2020 in 22 study centers in Germany. Participants were mainly recruited by case managers, company doctors, and flyers and posters that were provided by the central project management and customized by the company’s health insurance funds. Before participants were randomly assigned to one of the two study arms by case managers, company doctors and case managers decided which of the three modules would fit the employees’ needs most. Case managers were responsible for obtaining written informed consent [19].

This study focusses on employees in case management and module A for two reasons: (1) Study results do not allow us to comprehend treatment in the self-management group, and (2) the numbers of participants in modules B and C were too small for conducting regression analyses needed to answer the research questions.

### 2.1. Study Design and Participants

In module A, participants in case management care received an Evaluation of Functional Performance Capability (EFL). The results were used to set up a training schedule adjusted to workplace conditions which participants were supposed to follow two times a week in a training center and (partly) under the supervision of trainers for six weeks. The training schedule was adjusted by trainers when necessary. At the end of the intervention, participants received a second EFL to track if the training was successful. All participants were supported by a case manager during the program.

Participants received a paper-based questionnaire at the intervention’s beginning (t0) and six months later (t1). All participants with information on a disability score at t0 and t1, pain intensity at t0 and t1, psychological wellbeing at t0 and t1, motivation at t0, satisfaction at t1, training quality at t1, and age, gender, and educational degree at t0, were considered for analyses: a total of 180 participants remained (see Figure 1). Most participants in this study sample worked in the industrial sector (87.2%) with predominantly mental (47.2%) or equally mental and physical (35.0%) work tasks. For further information on the study sample, see also [19].

### 2.2. Measures

**Psychological wellbeing** was measured using the WHO-Five Well-Being Index [20]. The instrument consists of five questions regarding the last two weeks of the person, e.g., “Over the last two weeks I have felt cheerful and in good spirits,” that could be answered on a six-point scale from 0 (never) to 5 (the whole time). The raw value reaches from 0 (worst wellbeing) to 25 (best wellbeing). Values below 13 are seen as an indicator for depression screening [21]. In this study, Cronbach’s Alpha was 0.91 for t0 and 0.89 for t1. For our analyses, we calculated the difference between the t1 and t0 of psychological wellbeing_∆_.

**Pain intensity** was measured with excerpts of the German pain questionnaire [22]. In order to obtain a differentiated overview of the pain intensity, information was measured via three situation-related scales, asking for the current, the average, and the highest pain intensity within the last four weeks. The variable ranges from 0 to 100, whereby values below 50 are interpreted as low and values from 50 as high pain intensity [22]. For our analyses, we calculated the difference between the t0 and t1 of pain intensity_∆_.

In order to measure the actual state of experienced impairment due to pain, we used the **disability score** of the German pain questionnaire [22]. This dimension is of great prognostic importance and is very well-suited for demonstrating therapeutic efficiency. Effectiveness studies almost regularly show that the restoration of (experienced) active functioning is a necessary prerequisite for successful treatment. The scale records the extent to which the patient is impaired by his or her pain in everyday life, during leisure activities, and at work. The extent of pain-related, subjectively experienced impairment is assessed by the patient himself. The disability score is calculated by the mean of impairment in everyday life, leisure activity, and ability to work multiplied by ten (score values from 0 to 100) [22]. For our analyses, we also calculated the difference between the t0 and t1 of the disability score_∆_.

**Motivation** was measured on a scale reaching from 0 (worst conceivable motivation) to 10 (best conceivable motivation). **Satisfaction** was measured by asking for satisfaction with the measures and the program by their personal opinion of the program (1 very bad to 5 very good).

In order to check for quality characteristics of the training, the questionnaire included 3 items: a given training plan to follow; the possibility to approach a trainer at any time; regular appointments to adjust the training plan. Answers were coded with 1 = yes and 2 = no. The index **training quality** reached from 3 (best quality) to 6 (worse quality).

### 2.3. Statistical Analysis

We conducted paired *t*-tests and chi-square tests to test for (H1), change in psychological wellbeing from beginning to end of the intervention, and for (H2), differences in pain intensity (and further variables) of individuals with low and high psychological wellbeing.

Multiple linear regression analyses were used to test if (H3) the disability score mediates the impact of pain intensity on psychological wellbeing and if (H4) psychological wellbeing has an impact on the disability score.

**H3:** In model I, we used psychological wellbeing_t1_ as the dependent variable and pain intensity_∆_ as the independent variable. As confounding variables, we considered psychological wellbeing_t0_, motivation_t0_, satisfaction_t1_, training quality_t1_, and age, gender, and educational degree at t0 as the independent variables. In model II, we added the disability score_∆_ to test for mediation.**H4:** In model III, we used the disability score_t1_ as the dependent variable and psychological wellbeing_∆_ as the independent variable. As confounding variables, we considered the disability score_t0_, pain intensity_∆_, motivation_t0_, satisfaction_t1_, training quality_t1_, and age, gender, and educational degree at t0 as independent variables.

Before conducting multiple linear regression analyses, we checked for multi-collinearity (see Table A1 and Table A2). Our analyses were conducted using IBM SPSS Statistics 26.

## 3. Results

Age was measured in six categories (see Table 1). Of the study population, there were 75.9% male and 24.1% female participants. The main age groups were 40 to 49 and 50 to 59 years with 74.9%. Most individuals had at least a vocational degree (57.2%) or a completed education at, e.g., a technical school or vocational academy (24.1%). The mean of the disability score decreased from 40.33 (*SD* = 22.24) at t0 to 24.54 (*SD* = 21.23) at t1. Pain intensity also decreased from 49.52 (*SD* = 19.65) at t0 to 35.02 (*SD* = 20.85) at t1. Psychological wellbeing increased from 13.34 (*SD* = 5.09) at t0 to 15.09 (*SD* = 4.92) at t1 (see Table 1). Changes over time were significant for: the disability score with *t*(179) = 9.04, *p* < 0.001; pain intensity with *t*(179) = 9.92, *p* < 0.001; and psychological wellbeing, with *t*(179) = −4.65, *p* < 0.001.

*T*-tests and chi-square tests revealed significant differences in individuals with low and high psychological wellbeing for: the disability score_t0,t1_ with *t*_t0_(178) = −4.230, *p*_t0_ < 0.001 and *t*_t1_(178) = −2.733, *p*_t1_ < 0.001; pain intensity_t0,t1_ with *t*_t0_(178) = −3.127, *p*_t0_ < 0.01 and *t*_t1_(178) = −3.345, *p*_t1_ < 0.01; and motivation_t0_ with *t*_t0_(178) = 4.223, *p*_t0_ < 0.001 (see Table 2).

Model I showed significant results for the determinant factors of psychological wellbeing_t0_ (beta = 0.400, *p* < 0.001), pain intensity_∆_ (beta = 0.190, *p* < 0.05), motivation_t0_ (beta = 0.174, *p* < 0.05), and satisfaction_t1_ (beta = 0.263, *p* < 0.001). Training quality_t1_ (beta = −0.021, *p* > 0.05), age (beta = −0.039, *p* > 0.05), gender (beta = 0.006, *p* > 0.05), and educational degree (beta = 0.017, *p* > 0.05) had no impact on psychological wellbeing_t1_ in this model. Adjusted R^2^ was 0.412 (see Table 3).

Model II showed significant results for the determinant factors of psychological wellbeing_t0_ (beta = 0.419, *p* < 0.001), the disability score_∆_ (beta = 0.155, *p* < 0.05), motivation_t0_ (beta = 0.167, *p* < 0.05), and satisfaction_t1_ (beta = 0.227, *p* < 0.01). Pain intensity_∆_ (beta = 0.102, *p* > 0.05), training quality_t1_ (beta = −0.056, *p* > 0.05), age (beta = −0.023, *p* > 0.05), gender (beta = 0.019, *p* > 0.05), and educational degree (beta = 0.039, *p* > 0.05) had no impact on psychological wellbeing_t1_ in this model. Adjusted R^2^ was 0.422 (see Table 4).

The results of multiple linear regression analysis showed significant results for the determinant factors of the disability score_t0_ (beta = 0.568, *p* < 0.001), pain intensity_∆_ (beta = −0.462, *p* < 0.001), psychological wellbeing_∆_ (beta = −0.161, *p* < 0.01), satisfaction_t1_ (beta = −0.191, *p* < 0.01), and training quality_t1_ (beta = −0.131, *p* < 0.05) in model III. Age (beta = 0.101, *p* > 0.05), gender (beta = 0.065, *p* > 0.05), and educational degree (beta = 0.075, *p* > 0.05) had no impact on the disability score_t1_ in this model. Adjusted R^2^ was 0.517 (see Table 5).

## 4. Discussion

This study examines the role of psychological wellbeing in a WHC program for employees with MSDs. To summarize the findings:**H1:** T-tests’ results confirmed the hypothesis that the psychological wellbeing of employees significantly increased after the intervention.**H2:** Regarding differences in individuals with low and high psychological wellbeing_t0_, we found significant results for the disability score_t0,t1_, pain intensity_t0,t1_, and motivation_t1_. The disability score_t0,t1_ and pain intensity_t0,t1_ were higher in individuals with low psychological wellbeing_t0_; motivation_t0_ was higher in individuals with high psychological wellbeing_t0_.**H3:** Pain intensity_∆_ was not significant for psychological wellbeing_t1_ after adding the disability score_∆_ in model II. This indicates that, in our study sample, impairment due to pain totally mediates the impact of pain intensity on psychological wellbeing.**H4:** Results in model III indicate that the more the psychological wellbeing of participants increased, the better their disability score_t1_ was.

Additional analyses indicate that the disability score_t1_ increased with increasing quality of the training. The training aims at strengthening an individual’s musculoskeletal system according to the results of EFL. As the training quality includes a given training plan, the possibility to approach a trainer at any time, and regular appointments to adjust the training plan, this result was to be expected.

Psychological wellbeing_t1_ increased with higher levels of motivation_t0_. Highly motivated employees might have been particularly reliable in following their training schedule and feeling self-effective [23], which would have had positive effects on their psychological wellbeing_t1_.

Satisfaction_t1_ showed significant effects on both psychological wellbeing_t1_ and the disability score_t1_: the higher the satisfaction_t1_, the higher the psychological wellbeing_t1_ and the lower the disability score_t1_. The result may be based on the fact that individuals with higher psychological wellbeing and/or a lower disability score after intervention were likely to be satisfied.

Pain intensity_∆_ was significant for the disability score_t1_ in model I—as the disability score measures the experienced impairment due to pain, this result was to be expected.

The study results are congruent with, and contribute to, the current state of research and highlight the importance of psychological wellbeing in a worksite health promotion program for employees with first minor complaints regarding their musculoskeletal system. In this study, the intervention covered no specific aspects of psychological wellbeing. However, the psychological wellbeing of participants significantly increased from t0 to t1, which may indicate that the intervention does not only positively impact pain intensity and disability score but also psychological wellbeing. The causality of this finding should be addressed in future research, as the results may (also) be based on the positive changes in pain intensity and the disability score of individuals.

In Module A, the case management group had superior performance compared to the self-management group, as the case management group had fewer disability days, lower disability scores, lower pain intensity, higher self-efficacy values, and higher work ability at t1 than the self-management group. However, changes over time were significant not only in the case management group but also in the self -management group for disability days, disability scores, and work ability (see [19]). The results of this study supplement the original study results [19] by providing orientation for future adaption of Module A’s case management intervention. As the training in this WHC program was adapted to individuals’ workplaces, some aspects of work were already considered in the planning and execution of the intervention. Macdonald and Oakman [24] recommend focussing on psychosocial work stressors when planning interventions on MSDs at the workplace. The setting of this study holds potential, as participants did get into contact with case managers and/or company doctors. Because of the comorbidity with mental health problems, the approach via MSDs could be suitable to address the often still stigmatized topic of mental health via a low-threshold/more accepted approach. As psychosocial work factors not only have an impact on psychological wellbeing but also on MSD [2,3,4,5,6,7,8,9], the intervention could be improved by, e.g., adding modules that specifically focus on psychosocial work factors to improve psychological wellbeing. It might be helpful to not only identify individuals’ actual physical work situation but also to identify psychological wellbeing at the intervention’s beginning and, specifically, support employees with problematic values. Participants with low psychological wellbeing at the intervention’s beginning, in particular, have the potential to improve their psychological wellbeing, which also might have a positive impact on their disability score after the intervention. However, individuals started with inconspicuous values of psychological wellbeing_t0_ [21]. This generally implies including psychological aspects in interventions similar to the presented, as not only individuals with low psychological wellbeing may benefit but individuals in general.

### Strenghts and Limitations

A major strength of the study is that the recruitment and intervention took place in 22 study centres, covering 20 companies of different branches (industry, manufacturing, tourism, local governments, and consulting). Furthermore, the intervention was carried out in a “real-world setting,” with different enabling and hindering factors [19]. However, the RCT design might have had an effect on the selection of study participants.

The results may have been biased, as most of the companies of the study network (1) were very likely companies where employee health was already in focus before study participation and (2) already had contact with their company health insurance funds prior to study participation and were recruited to this study via the company health insurance funds.

In this study sample, mainly men and individuals aged between 40 and 59 years were represented. The age distribution is characteristic for individuals with MSDs and of working age. The higher proportions of men are likely to be attributed to the higher proportions of men in the participating industrial companies [25].

Motivation_t0_ in the study sample was relatively high, with a mean of 8.09 of 10. Participation was voluntary, and recruitment may only have reached those employees with already high motivation. As mentioned above, satisfaction_t1_ after the intervention was high. This might be due to the case that, in this study, the only results considered were those of individuals who answered both questionnaires and, therefore, likely finished the training. Individuals who were not satisfied with the program may have dropped out of the intervention and study.

## 5. Conclusions

The more the psychological wellbeing of participants in a WHC program for employees with first minor complaints in their musculoskeletal system improves, the lower their experienced impairment due to pain after an intervention is. Therefore, psychological wellbeing is not only of value for interventions that focus on individuals with severe or chronic musculoskeletal disorders but also for those addressing individuals from the perspective of prevention. The lower the psychological wellbeing is before an intervention, the higher the potential is for improvement in psychological wellbeing and, therefore, in individuals who experienced impairment due to pain. Interventions should include both pain-related and psychological aspects to achieve the best possible outcomes. However, the relationship between pain intensity, impairment due to pain, and psychological wellbeing is complex. Future research needs to explore the causality of pain intensity, impairment due to pain, and psychological wellbeing further in the context of workplace healthcare interventions with preventive character for individuals with first to slight discomforts in their musculoskeletal system.

## Figures and Tables

**Figure 1 ijerph-19-05452-f001:**
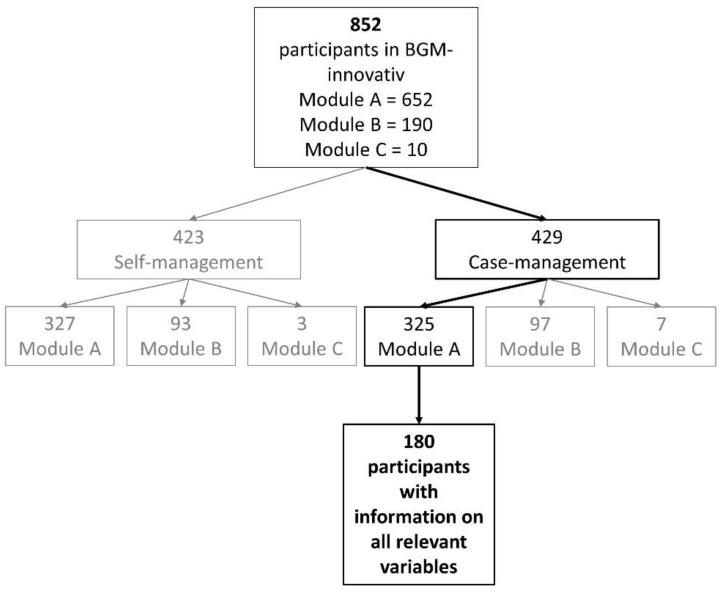
Flow chart of the study sample.

**Table 1 ijerph-19-05452-t001:** Descriptives of study population.

	N		%	
Age				
*≤29 years*	9		5.0	
*30 to 39 years*	23		12.8	
*40 to 49 years*	52		28.9	
*50 to 59 years*	81		45.0	
*60 to 69 years*	15		8.3	
*≥70 years*	0		0.00	
Gender				
*Male*	136		75.6	
*Female*	44		24.4	
Educational degree				
*No vocational degree*	8		4.4	
*Completed vocational education*	101		56.1	
*completed training at a technical school, master school, vocational academy, or technical academy*	45		25.0	
*College degree*	19		10.6	
*University degree*	7		3.9	
	**N**	**M**	**SD**	**Median (min/max)**
Disability score_t0_	180	40.33	22.24	40.00 (0.00/90.00)
Disability score_t1_	180	24.54	21.23	20.00 (0.00/100.00)
Pain intensity_t0_	180	49.52	19.65	50.00 (3.33/93.33)
Pain intensity_t1_	180	35.02	20.85	33.33 (0.00/93.33)
Psychological wellbeing_t0_	180	13.34	5.09	14.00 (5.00/24.00)
Psychological wellbeing_t1_	180	15.09	4.92	16.00 (1.00/25.00)
Motivation_t0_	180	8.09	1.49	8.00 (2.00/10.00)
Satisfaction_t1_	180	4.24	0.77	4.00 (1.00/5.00)
Training quality_t1_	180	3.28	0.74	3.00 (3.00/9.00)

Notes: M = mean value; SD = standard deviation.

**Table 2 ijerph-19-05452-t002:** Differences in individuals with low and high psychological wellbeing.

	Low Psychological Wellbeing_t0_ ^1^				High Psychological Wellbeing_t0_ ^1^				*t*-Test *p*-Value
	N	M	SD	Median	N	M	SD	Median	
Pain intensity_t0_	73	54.93	18.01	56.67	107	45.83	19.94	46.67	0.002 **
Pain intensity_t1_	73	41.14	20.66	40.00	107	30.84	20.02	30.00	0.001 **
Disability score_t0_	73	48.45	20.09	50.00	107	34.80	22.02	30.00	0.000 ***
Disability score_t1_	73	29.68	22.20	26.67	107	21.03	19.89	16.67	0.007 *
Training quality_t1_	73	3.34	0.67	3.00	107	3.23	0.78	3.00	0.334
Motivation_t0_	73	7.55	1.62	8.00	107	8.46	1.27	8.00	0.000 ***
Satisfaction_t1_	73	4.23	0.76	4.00	107	4.25	0.79	4.00	0.869
**Variable Response Trait**	**Low psychological wellbeing_t0_ ^1^** **(Percentage)**				**High psychological wellbeing_t0_ ^1^** **(Percentage)**				**Chi-Square Test** ***p*-value**
Gender									
*Men*	53 (29.4)				83 (46.1)				0.446
*Women*	20 (11.1)				24 (13.3)				
Age ^2^									
*≤29 to 49 years*	32 (17.8)				52 (28.9)				0.529
*50 to ≥* *70 years*	41 (22.8)				55 (30.6)				
Educational status ^2^									
*no vocational degree*/*completed vocational education*	44 (24.4)				65 (36.1)				0.949
*higher*	29 (16.1)				42 (23.3)				

Notes: M = mean value; SD = standard deviation. ^1^ Low psychological wellbeing (≥13); high psychological wellbeing (<13). ^2^ Age and educational degree were, due to small group sizes, considered as bivariate variables. * *p* < 0.05. ** *p* < 0.01. *** *p* < 0.001.

**Table 3 ijerph-19-05452-t003:** Multiple linear regression analysis, model I (dependent variable: psychological wellbeing_t1_).

Determinant Factors ^1^	Regression Coefficient B (SE)	Beta	*p*-Value	95% Confidence Interval		R^2^ (Adjusted)
				Lower Value	Upper Value	
Psychological wellbeing_t0_	0.387 (0.061)	0.400	<0.001 ***	0.267	0.507	0.438 (0.412)
Pain intensity_∆_	0.048 (0.015)	0.190	0.002 **	0.018	0.077	
Motivation_t0_	0.576 (0.224)	0.174	0.011 *	0.133	1.019	
Satisfaction_t1_	1.673 (0.425)	0.263	<0.001 ***	0.834	2.512	
Training quality_t1_	−0.140 (0.422)	−0.021	0.741	−0.972	0.692	
Gender	0.064 (0.668)	0.006	0.924	−1.255	1.382	
Age ^2^	−0.379 (0.577)	−0.039	0.512	−1.518	0.760	
Educational degree ^2^	0.173 (0.588)	0.017	0.769	−0.988	1.334	

Notes: SE = standard error. ^1^ Dependent variable: psychological wellbeing_t1_. ^2^ Age and educational degree were, due to small group sizes, considered as bivariate variables (age group 1 = ≤29 to 49 years and group 2 = 50 to ≥70 years; educational degree group 1 = no vocational degree and completed vocational education and group 2 = higher). * *p* < 0.05. ** *p* < 0.01. *** *p* < 0.001.

**Table 4 ijerph-19-05452-t004:** Multiple linear regression analysis, model II (dependent variable: psychological wellbeing_t1_).

Determinant Factors ^1^	Regression Coefficient B (SE)	Beta	*p*-Value	95% Confidence Interval		R^2^ (Adjusted)
				Lower Value	Upper Value	
Psychological wellbeing_t0_	0.406 (0.061)	0.419	0.000 ***	0.286	0.526	0.451 (0.422)
Pain intensity_∆_	0.026 (0.018)	0.102	0.166	−0.011	0.062	
Disability score_∆_	0.033 (0.016)	0.155	0.046 *	0.001	0.064	
Motivation_t0_	0.555 (0.223)	0.167	0.014 *	0.116	0.995	
Satisfaction_t1_	1.442 (0.437)	0.227	0.001 **	0.580	2.304	
Training quality_t1_	0.375 (0.434)	−0.056	0.389	−1.231	0.482	
Gender	0.217 (0.666)	0.019	0.745	−1.098	1.533	
Age ^2^	−0.228 (0.577)	−0.023	0.693	−1.367	0.911	
Educational degree ^2^	0.392 (0.593)	0.039	0.509	−0.779	1.563	

Notes: SE = standard error. ^1^ Dependent variable: psychological wellbeing_t1._
^2^ Age and educational degree were, due to small group sizes, considered as bivariate variables (age group 1 = ≤29 to 49 years and group 2 = 50 to ≥70 years; educational degree group 1 = no vocational degree and completed vocational education and group 2 = higher). * *p* < 0.05. ** *p* < 0.01. *** *p* < 0.001.

**Table 5 ijerph-19-05452-t005:** Multiple linear regression analysis, model III (dependent variable: disability score_t1_).

Determinant Factors ^1^	Regression Coefficient B (SE)	Beta	*p*-Value	95% Confidence Interval		R^2^ (Adjusted)
				Lower Value	Upper Value	
Disability score_t0_	0.543 (0.054)	0.568	<0.001 ***	0.435	0.650	0.542 (0.517)
Pain intensity_∆_	−0.499 (0.062)	−0.462	<0.001 ***	−0.622	−0.377	
Psychological wellbeing_∆_	−0.678 (0.237)	−0.161	0.005 **	−1.145	−0.211	
Motivation_t0_	−1.064 (0.818)	−0.074	0.195	−2.679	0.552	
Satisfaction_t1_	−5.227 (1.722)	−0.191	0.003 **	−8.627	−1.827	
Training quality_t1_	−3.754 (1.695)	−0.131	0.028 *	−7.100	−0.407	
Gender	3.206 (2.613)	0.065	0.222	−1.952	8.363	
Age ^2^	4.289 (2.249)	0.101	0.058	−0.149	8.728	
Educational degree ^2^	3.255 (2.335)	0.075	0.165	−1.355	7.865	

Notes: SE = standard error. ^1^ Dependent variable: disability score_t1._
^2^ Age and educational degree were, due to small group sizes, considered as bivariate variables (age group 1 = ≤29 to 49 years and group 2 = 50 to ≥70 years; educational degree group 1 = no vocational degree and completed vocational education and group 2 = higher). * *p* < 0.05. ** *p* < 0.01. *** *p* < 0.001.

## Data Availability

Data are available from the study group on reasonable request. Please contact the corresponding author.

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
