# Peer review of "The Role of Psychological Wellbeing in a Cross-Provider Worksite Healthcare Management Program for Employees with Musculoskeletal Disorders"

_ijerph, 2022, doi:10.3390/ijerph19095452_

Round 1

Reviewer 1 Report

This is an interesting study and it is of interest for experts in the occupational health and safety setting. However, I have some questions regarding certain definitions and methodological aspects:

  1. The authors mix up secondary and tertiary pervention as well as health promotion. Since an individual has symptoms, you are not located in the secondary prevention anymore. Secondary prevention is only carried out in asymptomatic individuals per definitionem. All modules are therefore tertiary prevention programs.
  2. Regarding psychological wellbeing:  Wellbeing is classical health promotion and needs to be distinguished from preventive measures who focus on pathogenesis. Mental disorders, as mentioned in the introduction, include major depression or schizophrenia - I am quite sure that this is not the authors focus. Back pain might often be a result of a psychosomatic disorder. The authors should differentiate between disorders and wellbeing and underline the focus of this analyses. 
  3. Where are the participants recruited from? Which companies? Which industrial sector? Are they mainly working physically or do they have an office job?
  4. What were the interventions in the respective modules? Although this is not the main focus of this project, a small insight on what the interventional study was dealing with would be helpful .
  5. Statistical analysis L177 (H3): What is the dependent variable here? 
  6. I assume, there might be some selection bisas regarding the recruited companies or employees. I wonder if companies who are already focused on the wellbeing of their employees were more likely to participate in such a program? thsi should be addressed in the limitations section.

Author Response

Dear Reviewer,

We appreciate your time and effort for the revision of our manuscript! Thank you very much for your supportive comments, that help us to improve the quality of our manuscript! Please find below our point-by-point answers to your comments.

Comment: The authors mix up secondary and tertiary pervention as well as health promotion. Since an individual has symptoms, you are not located in the secondary prevention anymore. Secondary prevention is only carried out in asymptomatic individuals per definitionem. All modules are therefore tertiary prevention programs.

Answer: We revised the terms in our manuscript and eliminated “secondary” prevention, see lines 290 and 395. Furthermore, it now says “workplace healthcare (WHC) program” instead of “workplace health promotion (WHP) program”, see lines 68, 72, 82, 290, 346 and 390.

Comment: Regarding psychological wellbeing: Wellbeing is classical health promotion and needs to be distinguished from preventive measures who focus on pathogenesis. Mental disorders, as mentioned in the introduction, include major depression or schizophrenia - I am quite sure that this is not the authors focus. Back pain might often be a result of a psychosomatic disorder. The authors should differentiate between disorders and wellbeing and underline the focus of this analyses.

Answer: Thank you very much for this comment. Your are right. To state more clearly that the focus of our manuscript is on psychological wellbeing and not on mental disorders, we revised the introduction section, by eliminating (some aspects in) lines 41 to 50. We have opted for this decision as this prevents the potential ambiguities on the manuscripts’ focus.

Comment: Where are the participants recruited from? Which companies? Which industrial sector? Are they mainly working physically or do they have an office job?

Answer: We have added this information (based on data possibilities), see lines 111 to 112 and 161 to 163.

Comment: What were the interventions in the respective modules? Although this is not the main focus of this project, a small insight on what the interventional study was dealing with would be helpful.

Answer: We have added this information, see lines 113 to 125.

Comment: Statistical analysis L177 (H3): What is the dependent variable here?

Answer: Thank you for your comment. This was a mistake, which we eliminated, see now line 208.

Comment: I assume, there might be some selection bisas regarding the recruited companies or employees. I wonder if companies who are already focused on the wellbeing of their employees were more likely to participate in such a program? thsi should be addressed in the limitations section.

Answer: This is right. We added this aspect in the strenghts and limitations section, see lines 374 to 378.

Reviewer 2 Report

Thank you for the opportunity to review this interesting paper regarding psychosocial wellbeing's role on the impact of a worksite health promotion program for musculoskeletal complaints. The article is well-organized and well-written. The authors use a small sub-section of participants from a randomized trial for the analysis. The analyses do not utilize the advantages of a randomized trial by including both arms of the study.

I have a few comments regarding the study, that I hope can improve the presentation of results.

1) Line 113: "Job is on danger" should be "job is in danger"
2) Materials and Methods: Rating this paper using the Cochrane Risk of Bias tool would be difficult because some information is not reported well. Very little information is given about the setting. Were participants only office workers? How were participants recruited? How were participants randomized and allocated to groups? The authors could consider offering more general information here instead of only referencing the original paper. 
3) What exactly was the difference between the case-management and self-management groups? How intense was case management? The fact that the self-management group was the control group is unclear in the methods.
4) Statistical Analysis, Line 171: Were paired t-tests used?
5) Statistical Analysis, Lines 177: two variables are listed as independent variables and no dependent variable is listed.
6) Results, Tables 3-5: It is hard to determine the outcome (dependent variable) for these regression models. The dependent variable should be listed in the table descriptions.   
7)The exciting results from this study would be to see if the improvements seen in the intervention group (case management, module A) were greater after six months than in the control group (self-management, module A). I assume the authors made no comparison here because this was in the original paper [26]. Maybe authors can discuss the main results of the study briefly in the discussion and explain how these results supplement the original study results.
8) Line 43-44: The word "early" needs to come later in the sentence... or the sentence rewritten to talk about "early identification of employees."  

Author Response

Dear Reviewer,

We appreciate your time and effort for the revision of our manuscript! Thank you very much for your supportive comments, that help us to improve the quality of our manuscript! Please find below our point-by-point answers to your comments.

Comment: Line 113: "Job is on danger" should be "job is in danger"

Answer: Thank you for your comment. We revised this, see now line 118.

Comment: Materials and Methods: Rating this paper using the Cochrane Risk of Bias tool would be difficult because some information is not reported well. Very little information is given about the setting. Were participants only office workers? How were participants recruited? How were participants randomized and allocated to groups? The authors could consider offering more general information here instead of only referencing the original paper.

Answer: Thank you for that comment. We added more information on these aspects, see lines 111 to 132 and lines 161 to 163.

Comment: What exactly was the difference between the case-management and self-management groups? How intense was case management? The fact that the self-management group was the control group is unclear in the methods.

Answer: We added this information, see lines 113 to 125.

Comment: Statistical Analysis, Line 171: Were paired t-tests used?

Answer: Yes, we added this information, see now line 201.

Comment: Statistical Analysis, Lines 177: two variables are listed as independent variables and no dependent variable is listed.

Answer: Thank you for your comment. This was a mistake, which we eliminated, see now line 208.

Comment: Results, Tables 3-5: It is hard to determine the outcome (dependent variable) for these regression models. The dependent variable should be listed in the table descriptions.

Answer: Thank you very much for noticing. We revised the table description of Table 3 (see line 255) and added information on the dependent variable in Tables 3-5 (see lines 253, 269 and 283).

Comment: The exciting results from this study would be to see if the improvements seen in the intervention group (case management, module A) were greater after six months than in the control group (self-management, module A). I assume the authors made no comparison here because this was in the original paper [26]. Maybe authors can discuss the main results of the study briefly in the discussion and explain how these results supplement the original study results.

Answer: Thank you for your comment. You are right. We added more information on this aspect in the discussion section, see lines 338 to 345.

Comment: Line 43-44: The word "early" needs to come later in the sentence... or the sentence rewritten to talk about "early identification of employees."  

Answer: Thank you for your comment. This sentence is not longer part of the introduction section due to our revision based on the comments of the second reviewer, see lines 45 to 47.

Round 2

Reviewer 1 Report

All my comments were sufficiently adressed. I recommend to publish the manuscript in its current form.